# Hierarchical folding-upon-binding of an intrinsically disordered protein

Lenette F. Kjaer [1], Francesco S. Ielasi [2,4], Thomas Winbolt [1], Elise Delaforge [1], Maud Tengo[1], Luiza Mamigonian Bessa[1,5], Laura Mariño Pérez [1,6], Elisabetta Boeri Erba [1], Guillaume Bouvignies [3] ✉, Andrés Palencia [2] ✉ & Malene Ringkjøbing Jensen [1] ✉

Intrinsically disordered proteins (IDPs) often undergo folding-upon-binding to their partners via short linear motifs, typically 5-15 amino acids in length. However, a significant proportion of IDPs do not adhere to this paradigm but fold upon binding through extended regions comprising multiple molecular recognition elements. For these IDPs, the binding mechanisms and the structural characteristics of their folding intermediates remain poorly understood. Here we unveil hierarchical folding of an IDP as it binds to its partner, exemplified by the disordered signaling effector POSH and the small GTPase Rac1. By combining nuclear magnetic resonance (NMR) spectroscopy and X-ray crystallography, we resolve at atomic resolution how POSH transitions from a fully disordered state to a highly ordered, Rac1-bound conformation through two structurally distinct folding intermediates. The folding of each element is contingent on the successful structuring of the preceding element, highlighting a hierarchical folding-upon-binding mechanism. Our work highlights the potential of targeting folding intermediates and conformational transitions to unlock therapeutic opportunities for IDPs.

Intrinsically disordered proteins (IDPs) make up approximately 40% of the human proteome and play essential roles in diverse biological processes such as signal transduction, transcriptional regulation, and cell cycle control[1,2]. Their misregulation has been linked to various diseases, including cancer and neurodegenerative disorders, positioning IDPs as promising next-generation drug targets[3-6]. Unlike structured proteins, IDPs do not adopt stable three-dimensional conformations, but instead exist as dynamic ensembles of interconverting states[7-10]. Many IDPs interact with their binding partners through short linear motifs (SLiMs), which are sequence segments typically 5-15 amino acids in length[11]. These motifs often undergo coupled folding and binding, adopting a defined structure upon partner association[12,13]. Experimental and computational studies have shed light on mechanisms underlying SLiM-mediated interactions, emphasizing the role of secondary structure preformation and transient encounter states in facilitating complex formation, often with fast association rates[14-20]. However, a significant proportion of IDPs do not adhere to this SLiM paradigm. Some exploit much longer interaction segments spanning multiple molecular recognition elements. The precise binding trajectory, the structural characteristics of potential folding intermediates and the thermodynamic forces driving such extensive interactions remain highly elusive.

[1]Université Grenoble Alpes, CNRS, CEA, IBS, Grenoble, France. [2]Institute for Advanced Biosciences (IAB), Structural Biology of Novel Targets in Human Diseases, INSERM U1209, CNRS UMR5309, Université Grenoble Alpes, Grenoble, France. [3]Chimie Physique et Chimie du Vivant (CPCV), Département de Chimie, École Normale Supérieure, PSL University, Sorbonne Université, CNRS, Paris, France. [4]Present address: Amoéba, Chassieu, France. [5]Present address: Evotec (France) SAS, Toulouse, France. [6]Present address: Departament de Química, Universitat de les Illes Balears, Institut Universitari d'Investigació en Ciències de la Salut (IUNICS), Institut de Recerca en Ciències de la Salut (IdISBa), Palma, Spain. ✉e-mail: guillaume.bouvignies@ens.psl.eu; andres.palencia@inserm.fr; malene.jensen@ibs.fr

Here we unveil hierarchical folding of an IDP as it binds to its partner, exemplified by the disordered signaling effector POSH and the small GTPase Rac1. We show that POSH lacks discernible secondary structure in its unbound state but adopts a distinct effector fold upon interaction with Rac1. Using nuclear magnetic resonance (NMR) spectroscopy, we demonstrate that POSH undergoes extensive folding-upon-binding, marked by sequential conformational transitions occurring over a time scale of seconds. Binding is initiated by specific anchoring contacts with Rac1, followed by the structuring of a first molecular recognition element. In this intermediate state, the remaining POSH sequence collapses while exploring binding-competent conformations, ultimately leading to the folding of a second molecular recognition element on the surface of Rac1. The folding of the second element is contingent on the successful structuring of the first element, underscoring the hierarchical nature of the process. By dissecting the structural details of this folding trajectory at atomic resolution, our study offers insight into binding-induced folding of long disordered sequences and highlights the potential of targeting folding intermediates and conformational transitions to unlock therapeutic opportunities for IDPs.

## Results

### POSH contains only a partial CRIB motif

The small GTPase Rac1 functions as a molecular switch, cycling between an inactive, guanosine diphosphate (GDP)-bound state and an active guanosine triphosphate (GTP)-bound state[21]. This transition involves structural rearrangements in the switch I loop shifting from a predominantly open conformation in the GDP-bound form to a primarily closed conformation in the GTP-bound form (Supplementary Fig. 1a−d)[22,23]. In response to extracellular stimuli, Rac1 interacts with effector proteins that coordinate the activation of signaling pathways to elicit tailored cellular responses[24]. Rac1 recognizes Cdc42/Rac1-interactive binding (CRIB) motifs within its effectors through a hydrophobic surface that emerges upon switch I loop closure, ensuring that only the active, GTP-bound form can engage effectors with high affinity[25,26]. One of these effectors is Plenty Of SH3s (POSH), a scaffold protein that mediates c-Jun N-terminal kinase (JNK) activation in apoptosis[27,28]. The Rac1-binding region of POSH has been mapped to the disordered region between its second and third SH3 domains (residues 260−445, Fig. 1a)[29]. However, this region contains only a partial CRIB motif (Fig. 1b, Supplementary Fig. 1e, f), prompting us to further investigate the Rac1-binding mode of POSH.

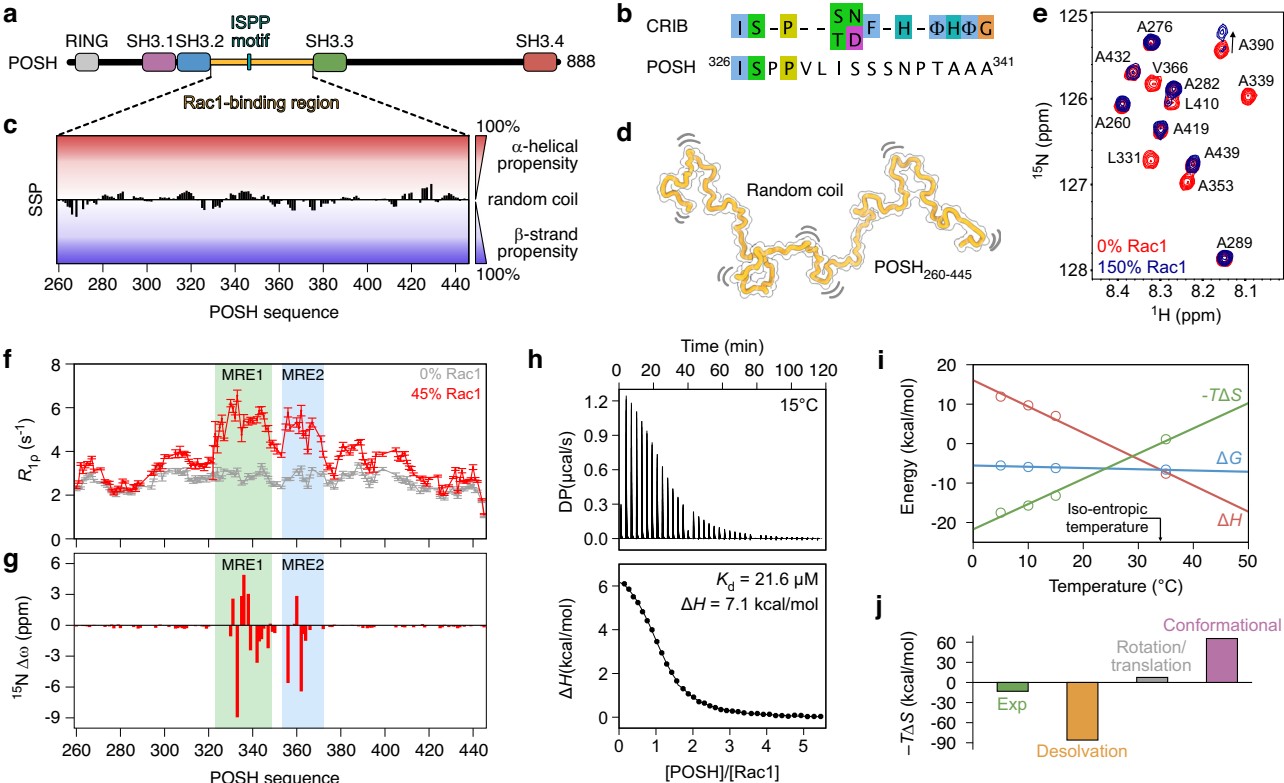

**Fig. 1 | POSH interacts with Rac1 using two molecular recognition elements that fold upon binding. a** Domain organization of POSH with the Rac1-binding region shown in orange. **b** Comparison of the canonical CRIB recognition motif with the sequence of POSH. Only three out of eleven canonical positions are conserved in POSH. **c** Secondary structure propensities of the Rac1-binding region of POSH calculated from experimental Cα and Cβ chemical shifts (black bars). **d** POSH$_{260-445}$ adopts an ensemble of random coil conformations, here represented by a single coil conformation undergoing dynamic fluctuations. **e** Zoom on a region of the $^1$H-$^{15}$N HSQC spectrum of POSH$_{260-445}$ without (red) and with (blue) 150% (molar ratio) of Rac1. **f** $^{15}$N $R_{1\rho}$ relaxation rates of POSH$_{260-445}$ without (gray) and with (red) 45% (molar ratio) of Rac1. The data were acquired at 25 °C and a $^1$H frequency of 700 MHz. Error bars represent one standard deviation of the relaxation rates obtained from Monte Carlo resampling of the intensity decays (see "Methods"). **g** Chemical shift differences (Δω) between the major (free) and minor

(Rac1-bound) state of POSH as observed in $^{15}$N CEST experiments acquired on $^{15}$N-labeled POSH with 20% (molar ratio) of Rac1 using a $B_1$ field of 21.6 Hz. **h** ITC titration of POSH$_{315-380}$ with Rac1. ITC data from three injections were merged by concatenation. Raw injection heats (top) and the corresponding specific binding isotherms (bottom) are shown. The data were acquired at 15 °C and analyzed according to the binding model "One set of sites" using the PEAQ-ITC analysis software (full-drawn line). **i** Temperature dependence of the thermodynamic parameters derived from ITC titrations of Rac1 with POSH$_{315-380}$. Points represent the average of experimental data from duplicate experiments and lines correspond to linear least-squares fits. **j** Dissection of the experimental binding entropy (green) obtained by ITC titrations of Rac1 with POSH$_{315-380}$ at 15 °C into contributions from desolvation of protein surfaces (orange), rotational and translational motions (gray) and conformational changes (magenta) according to a Spolar-Record analysis (see "Methods"). Source data are provided as a Source data file.

## Rac1 engages an extensive 50-residue region of POSH

NMR spectroscopy of the Rac1-binding region of POSH reveals a complete lack of discernible secondary structure (Fig. 1c, Supplementary Fig. 2a), showing that this region is best characterized by a dynamic ensemble of structures adopting random coil conformations (Fig. 1d). To produce the effector-binding capable form of Rac1, we exchanged GDP for a non-hydrolysable GTP analog, GMPPNP (Supplementary Fig. 2b). We then investigated the interaction between [15]N-labeled POSH and Rac1 using NMR titrations, revealing that the interaction occurs in the slow exchange regime on the NMR chemical shift time scale (Fig. 1e, Supplementary Fig. 2c, d). Notably, most peaks corresponding to residues within the direct interaction site of POSH disappear, however, no new peaks appear in the NMR spectra corresponding to the Rac1-bound state of POSH. This is most likely due to POSH remaining dynamic on the surface of Rac1.

To delineate the interaction site of Rac1, we measured [15]N $R_{1\rho}$ relaxation rates of POSH in the absence and presence of a 45% molar ratio of Rac1 (Fig. 1f) and determined the chemical shift differences between the free and bound state of POSH using [15]N chemical exchange saturation transfer (CEST) experiments (Fig. 1g, Supplementary Fig. 2e). These data reveal that Rac1 engages an extensive 50-residue region of POSH (residues 322–372), encompassing two distinct molecular recognition elements (MRE1 and MRE2) separated by a flexible linker (Fig. 1f, g). We also acquired [13]C′ CEST experiments on a shorter construct of POSH comprising both MRE1 and MRE2 (residues 315–380, denoted POSH$_{315-380}$) in the presence of Rac1 at a 20% molar ratio, with the aim of probing secondary structure changes upon binding. The observed [13]C′ chemical shift differences between the free and Rac1-bound state of POSH are consistent with folding-upon-binding. Notably, the [13]C′ chemical shifts suggest that MRE1 folds into a β-strand followed by an α-helix, whereas MRE2 adopts β-strand conformations (Supplementary Fig. 3).

## Desolvation of protein surfaces compensates for the entropic cost of folding

To investigate the thermodynamic forces driving the complex formation between POSH$_{315-380}$ and Rac1, we utilized isothermal titration calorimetry (ITC) experiments and their dependence on temperature (Fig. 1h, i, Supplementary Fig. 4, Supplementary Table 1). The experiments reveal a moderate binding affinity, varying from 21 to 46 μM between 35 °C and 5 °C, and a significant change in heat capacity ($\Delta C_p = -2.8$ kJ·mol$^{-1}$·K$^{-1}$, Fig. 1i). This suggests that POSH undergoes folding upon binding to Rac1, involving substantial burial of hydrophobic surface area. Using the Spolar-Record analysis[30] and its recent adaptation to complexes involving IDPs[31], we further dissected the entropic contribution into three components: conformational changes, rotational and translational motions and desolvation of protein surfaces (Fig. 1j, see "Methods"). Our results show that the entropic penalty associated with folding-upon-binding of POSH is compensated by desolvation of protein surfaces, resulting in a net favorable, entropic contribution to binding. Assuming an average per-residue loss of conformational entropy of −24 J·mol$^{-1}$·K$^{-1}$[30-32], we estimate that approximately 40 residues of POSH fold upon binding to Rac1, in excellent agreement with the size of the binding region determined by NMR (Fig. 1f, g). Together, our ITC experiments suggest that POSH undergoes substantial folding-upon-binding, driven primarily by desolvation of hydrophobic surfaces.

## POSH adopts a belt-like conformation upon binding to Rac1

To elucidate the structural basis of the interaction, we determined the crystal structure of Rac1 in complex with POSH at 1.25 Å resolution using a fusion construct linking the C-terminus of Rac1 to the N-terminus of POSH (Supplementary Table 2). The structure reveals extensive folding-upon-binding, with POSH adopting a belt-like conformation that wraps around the GTPase (Fig. 2a–c), burying an area of

1830 Å$^2$ of Rac1 which corresponds to 20% of its total surface area. Consistent with the NMR and ITC data, POSH binds to Rac1 using two molecular recognition elements spanning over 50 residues, both of which are well-defined in the electron density map (Fig. 2d, e). MRE1 adopts an extended conformation followed by a short β-strand and an α-helix, while MRE2 forms a β-hairpin structure, in agreement with our [13]C′ CEST data (Supplementary Fig. 3). Compared to previously characterized GTPase-effector complexes (Supplementary Fig. 5)[33-36], POSH exhibits an entirely different fold, with the α-helix within MRE1 and the β-hairpin within MRE2 representing structural features not observed in other bound effectors. To verify that the fusion construct does not alter the bound conformation of POSH, we also determined the crystal structure of Rac1 in complex with an unfused MRE1 peptide of POSH at 1.85 Å resolution (Supplementary Table 2). The nearly identical conformation of MRE1 in both structures confirms the validity of the fusion strategy (Supplementary Fig. 6).

The crystal structure of the POSH-Rac1 complex reveals that POSH interacts with two functionally critical regions of the GTPase (Fig. 2f). The switch I loop of Rac1 assumes a closed conformation, stabilized by hydrogen bonds between the third phosphate of GMPPNP and both the backbone amide of T35 and the side chain hydroxyl group of Y32. This closed conformation of switch I allows the binding of the MRE1 helix on top of the loop. Additionally, MRE2 partially covers the Switch II region, which serves as the primary binding site for guanine nucleotide exchange factors and is a key determinant of the catalytic activity of Rac1 (Fig. 2f). Notably, MRE2 also partially overlaps with MRE1, forming a backbone-backbone hydrogen bond between S355 (MRE2) and S334 (MRE1), as well as a side-chain interaction between S355 and S333 via an intercalated water molecule (Fig. 2g). These structural insights indicate that POSH stabilizes Rac1 in an active conformation and suggest that MRE2 binding requires MRE1 to adopt a folded state, with binding occurring either concurrently with, or subsequent to, MRE1 folding.

Closer examination of the structural details reveals that the complex is stabilized predominantly by hydrophobic interactions, along with a total of 25 hydrogen bonds reinforcing the Rac1-POSH interface. At the N-terminus of POSH, the partial CRIB motif ([326]ISPP[329]) within MRE1 occupies a narrow hydrophobic cavity on Rac1 (Fig. 2d, h, i). Specifically, the side chains of I326 and P329 engage with this cavity (Fig. 2i), while the extended conformation of the partial CRIB motif is stabilized by multiple hydrogen bonds between the backbone of POSH and the side chains of Rac1, including Y23, D47, K166, D170 and R174 (Fig. 2h). These interactions occur either directly or through intercalated water molecules at the binding interface. Beyond the partial CRIB motif, MRE1 extends into a short, anti-parallel β-strand, augmenting the central β-sheet core of Rac1, followed by an α-helix that loops back onto the β-strand (Fig. 2j). The α-helix engages in hydrophobic contacts with the Rac1 surface, involving the side chains of L331, P337, A339, A340 and I343 of POSH (Fig. 2k). MRE1 is connected to MRE2 via a flexible loop (residues 346–352) that is not well-defined in the electron density map. MRE2 adopts a β-hairpin conformation, with its interaction stabilized by hydrophobic contacts with Rac1, mediated by the side chains of V357, I359, L364, V366 of POSH (Fig. 2l, m). Interestingly, the artificial intelligence-based predictor AlphaFold2 accurately captures the structural details of MRE1, however, it fails to predict the β-hairpin conformation of MRE2 (Supplementary Fig. 7a). This limitation is even more pronounced in predictions made by AlphaFold3, which poorly models the structures of both MRE1 and MRE2 (Supplementary Fig. 7b). These observations underscore the continued importance of experimental structure determination of complexes involving IDPs.

In summary, our crystal structure reveals how POSH folds upon binding, wrapping around the GTPase in a conformation stabilized by multiple hydrogen bonds and hydrophobic interactions. The two molecular recognition elements of POSH adopt a fold not previously

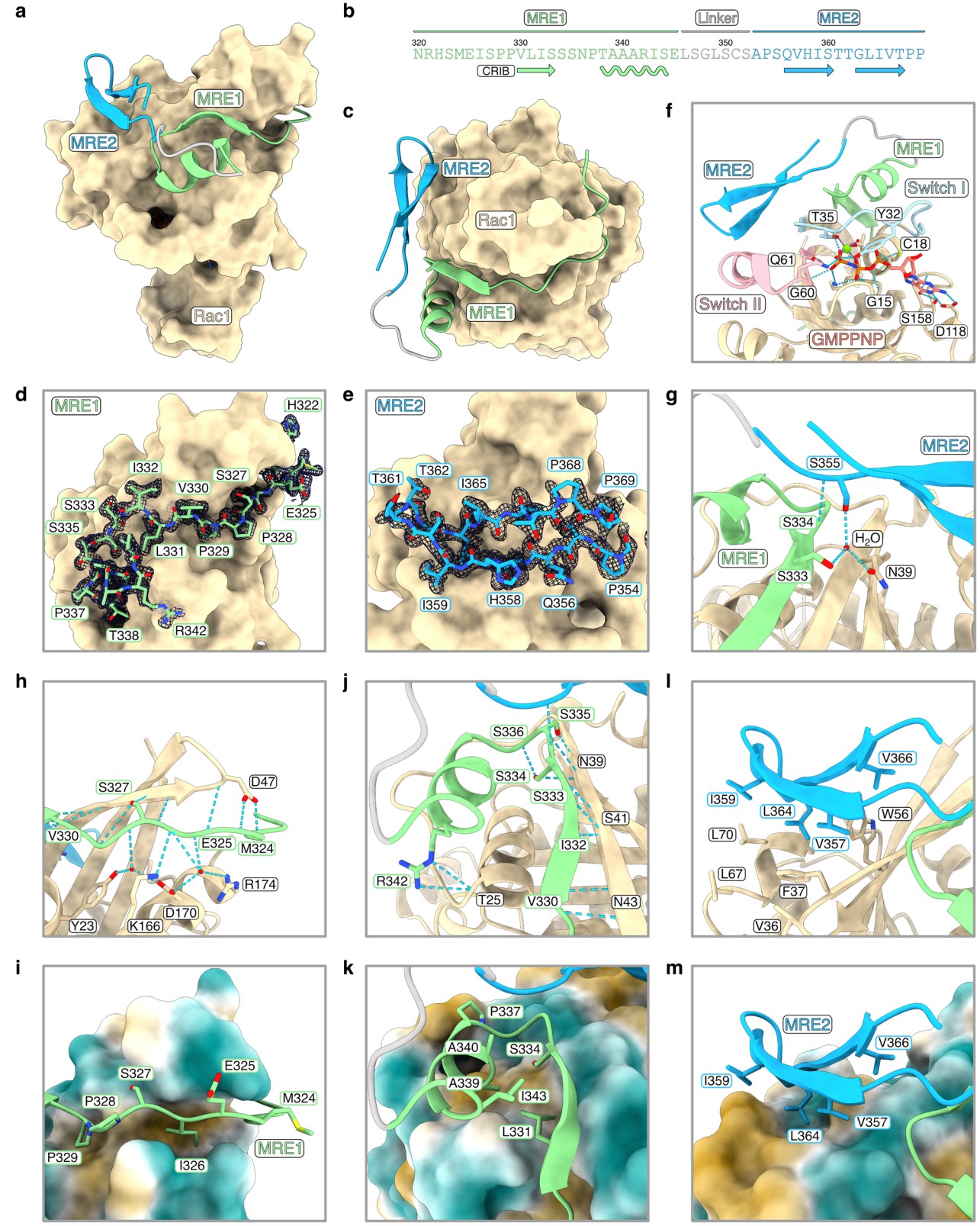

described for signaling effectors bound to the Cdc42/Rac GTPase family.

**POSH undergoes hierarchical folding upon binding to Rac1**

To uncover how POSH transitions from a completely disordered state in its unbound form to a structured conformation with multiple secondary structure elements, we initially investigated the interdependence of its two molecular recognition elements. To this end, we measured $^{15}$N $R_{1\rho}$ relaxation of POSH$_{315-380}$ at varying molar ratios of Rac1, observing modest increases in relaxation rates consistent with a slow apparent on-rate of the complex (Fig. 3a, Supplementary Fig. 8a, b). Mutation of the partial CRIB motif ($^{326}$ISPP$^{329}$ → $^{326}$RRPR$^{329}$) abolishes binding of all elements (Fig. 3b, Supplementary Fig. 8c), demonstrating that this motif is essential for binding and that

**Fig. 2 | High-resolution crystal structure of the Rac1-POSH complex. a** Overall structure of the Rac1-POSH complex. Rac1 is shown as a beige surface, while POSH is shown in green (MRE1), gray (linker) and blue (MRE2). The linker region, not well-defined in the electron density map, was modeled as a loop for clarity in all figures. **b** Sequence of POSH highlighting the location of MRE1 and MRE2 along with the secondary structure elements they adopt in the Rac1-bound state. The position of the partial CRIB motif ($^{326}$ISPP$^{329}$) is indicated. **c** Top-down view of the Rac1-POSH complex. **d** Composite omit electron density map of MRE1 contoured at 1.2σ, calculated using the CCP4 COMIT program[67]. **e** Composite omit electron density map of MRE2 contoured at 1.2σ. **f** Detailed view of the Rac1-POSH complex showing hydrogen bonds (dashed cyan lines) between Rac1 and GMPPNP (depicted in salmon and orange). Both switch I (light blue) and switch II (light pink) form hydrogen bonds with GMPPNP. The switch I loop adopts a closed conformation, enabling

MRE1 binding on top of this loop, while MRE2 partially covers the switch II region. **g** MRE2 folds back onto MRE1. Hydrogen bonds are established between the two elements. **h** Structure of the Rac1-POSH complex highlighting hydrogen bonds between Rac1 and the partial CRIB motif. **i** Close-up view of the partial CRIB motif with the Rac1 surface colored by hydrophobicity (ranging from dark orange for the most hydrophobic potentials, through white, to teal for the most hydrophilic potentials). **j**, Structure of the Rac1-POSH complex showing hydrogen bonds between Rac1 and MRE1. **k** Close-up view of MRE1 of POSH with the Rac1 surface colored according to hydrophobicity. **l** Structure of the Rac1-POSH complex showing that hydrophobic side chains mediate the interaction between Rac1 and MRE2. **m** Close-up view of MRE2 of POSH with the Rac1 surface colored by hydrophobicity.

MRE2 cannot interact with Rac1 independently, suggesting that MRE1 folding is required for MRE2 engagement. This observation is in agreement with our crystal structure, which shows MRE2 partially packing on top of MRE1 (Fig. 2g). Collectively, these results are compatible with a hierarchical folding mechanism of POSH, although they do not distinguish between sequential and concomitant folding of MRE1 and MRE2.

To address this question and gain insight into the kinetics of folding, we performed $^{15}$N CEST experiments on POSH with a 20% molar ratio of Rac1 at four different $B_1$ field strengths (5.2, 10.3, 20.5, and 68.6 Hz) in order to detect the NMR-invisible Rac1-bound states of POSH. The results support a slow, hierarchical folding mechanism, revealing two distinct bound states for multiple residues, particularly within MRE2. We analyzed the data using both a linear 3-site exchange model, in which MRE1 binds and folds simultaneously prior to MRE2 folding, and a 4-site exchange model, in which the partial CRIB motif binds first, followed by ordered sequential folding of MRE1 and MRE2 (Fig. 3c). To ensure robust data analysis, we fixed the total POSH-bound population at a value derived from the complex dissociation constant determined at 25 °C ($K_d = 24$ μM). Initial analysis of the data using the 3-site model was performed for each residue individually to assign minor dips in the CEST profiles to either the folding intermediate (B) or the fully complexed state (C) based on their populations (Supplementary Fig. 9). Subsequently, we performed a global fit of the CEST data across all residues using the same model. Although the 3-site model provided a reasonable fit to the experimental data, residues within the CRIB motif and MRE2 consistently showed deviations (Fig. 3d, Supplementary Fig. 10). To address this, we used a 4-site model (Fig. 3e, Supplementary Fig. 11), which showed a statistically significantly better agreement with the experimental data than the 3-site model (Supplementary Fig. 12a, b), and yielded a reliable determination of the exchange parameters, as confirmed by extensive grid searches (Supplementary Fig. 12c, d, Supplementary Table 3). Notably, the 4-site model explains all CEST data across the POSH sequence, eliminating the need for a clustering analysis in which data from different IDP segments are analyzed independently[14].

The analysis of the CEST data using the 4-site model provides a comprehensive view of the structural and kinetic details underlying the hierarchical folding mechanism of POSH. Folding initiates with the formation of a CRIB-anchored state (intermediate A), where the partial CRIB motif binds and the small β-strand of MRE1 is formed, while the remaining portion of MRE1 and MRE2 remain disordered, as evidenced by the chemical shift differences between the intermediate A and the free state F ($\Delta\omega_{FA}$, Fig. 3f). This first interaction step is characterized by both slow association and dissociation rates ($k_{on} = 7.3 \times 10^5$ M$^{-1}$s$^{-1}$ and $k_{off} = 56$ s$^{-1}$). The existence of intermediate A may be the result of the intrinsic dynamics of the switch I region of Rac1, which exchanges between a closed (major) and an open (minor) state[37]. While the CRIB-anchored state appears to be able to form regardless of the switch I conformation, an open switch I conformation (as observed in GDP-loaded Rac1, PDB 5N6O) would prevent folding of the α-helix of MRE1

due to steric clashes (Fig. 3i, j). Following CRIB-anchoring, the α-helix of MRE1 undergoes folding on the seconds time scale ($k_{AB} = 72$ s$^{-1}$ and $k_{BA} = 64$ s$^{-1}$). Surprisingly, MRE2 does not remain fully disordered in this intermediate state B. Instead, it exhibits significant chemical shift perturbations (Fig. 3g), suggesting that MRE2 collapses onto the surface of Rac1 while exploring binding-competent conformations. These conformations likely involve the formation of native-like contacts between MRE2 and the hydrophobic surface of Rac1, as evidenced by the MRE2 resonances in this intermediate state consistently appearing between those of the free (F) and fully complexed (C) states (Fig. 3e, g, h). This dynamic sampling of binding-competent conformations ultimately facilitates the folding of MRE2 into its β-hairpin structure ($k_{BC} = 89$ s$^{-1}$ and $k_{CB} = 94$ s$^{-1}$) completing the hierarchical folding trajectory of POSH (Fig. 3k, Supplementary Movie 1).

## Discussion

Our results reveal how POSH transitions from a fully disordered state to a folded structure in complex with Rac1, adopting multiple secondary structure elements. The identified hierarchical folding mechanism distinguishes itself from sequential binding models, where different motifs of an IDP can engage with a partner in any order[32], and from interaction mechanisms that rely on transient encounters before evolving to a fully bound state[14,16]. Notably, this mechanism enforces a strict order of molecular interactions with a key feature lying in its exclusivity. No other molecular recognition element of POSH can interact with Rac1 before transitioning through the CRIB-anchored state. This discovery not only deepens our understanding of the fascinating molecular recognition mechanisms of IDPs, but also suggests that hierarchical folding could play a key role in regulation of IDP function, particularly in signal transduction and molecular assembly, where binding dynamics play a crucial role.

The observed folding-upon-binding mechanism resonates with general principles of GTPase-effector recognition, including previously described dock-and-coalesce models. In these models, initial recognition of CRIB motif residues is followed by a staged collapse into the fully bound state[38–40]. Our results further complete this model by showing that the order of folding events can be strictly enforced, suggesting that this hierarchical folding-upon-binding may represent a broader paradigm for GTPase-effector interactions.

Placing our findings in the context of other GTPase-effector complexes reveals some differences with canonical systems. The binding affinity of POSH for Rac1 ($K_d = 24$ μM) is markedly weaker than that observed for other GTPase-effector complexes, including Cdc42-ACK ($K_d = 770$ nM), Cdc42-PAK1 ($K_d = 620$ nM), and Cdc42-WASP ($K_d = 55$ nM)[41]. One plausible explanation for this affinity difference is that POSH contains a non-canonical CRIB motif, which lacks several of the hydrophobic residues that normally serve as anchoring points for GTPase recognition and stabilization. Notably, in canonical CRIB-containing effectors, mutation of these conserved hydrophobic positions typically reduces GTPase affinity by up to ~100-fold, shifting dissociation constants into the micromolar

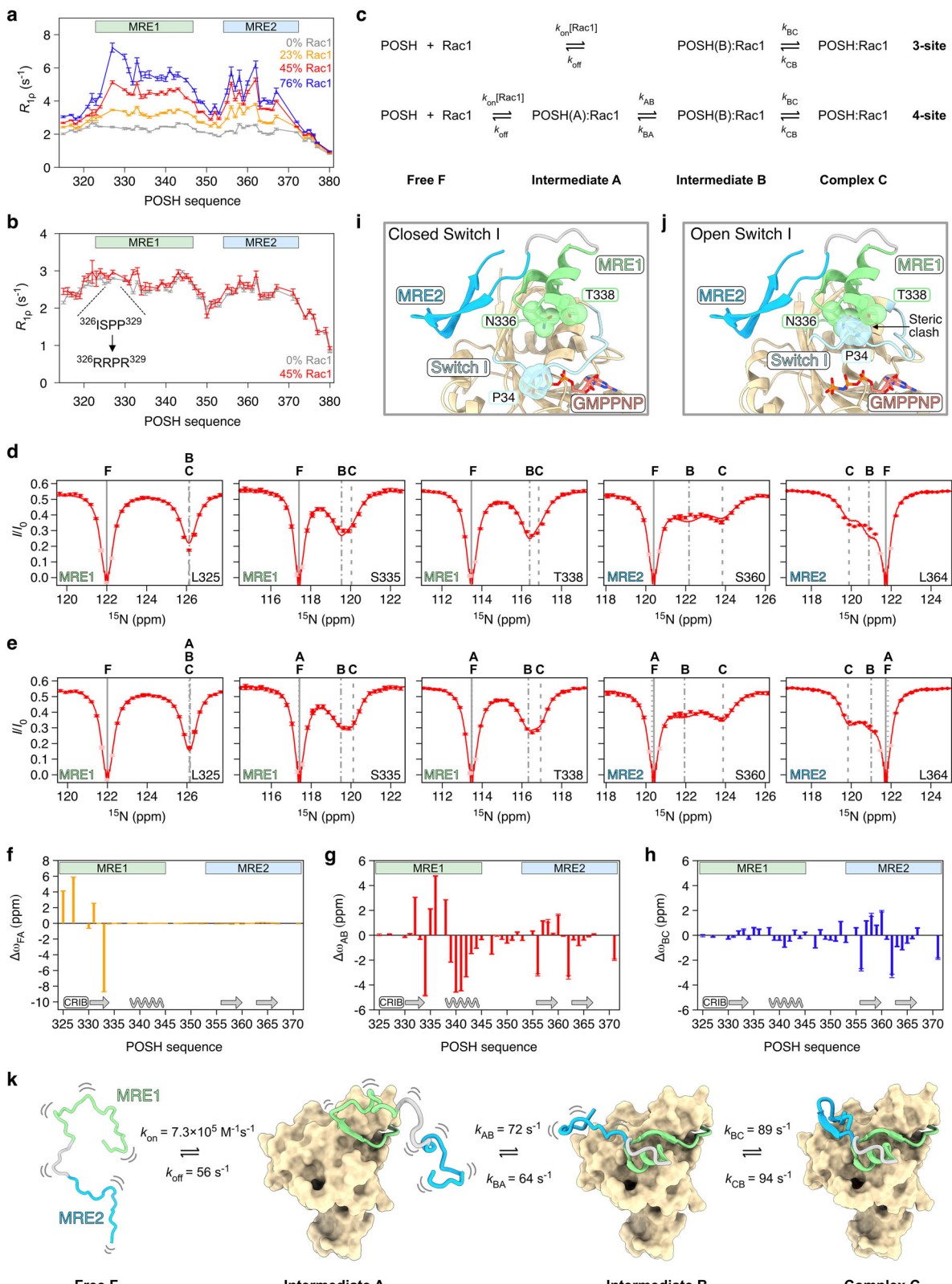

range[39]. Furthermore, our ITC experiments show that the extensive folding of POSH upon Rac1 binding imposes a considerable conformational entropy cost. Although compensated by the favorable release of water molecules from the protein interfaces, this entropic penalty may further weaken the overall binding affinity. This weaker, non-canonical POSH-Rac1 interaction likely reflects a functional adaptation compared to canonical interactions. Beyond the CRIB motif, POSH harbors multiple functional domains mediating protein-protein interactions, and transient, reversible binding to Rac1 may facilitate dynamic regulation of POSH without prolonged sequestration of the GTPase[27,28]. In this context, Rac1 binding may represent one component of a broader regulatory mechanism in

**Fig. 3 | POSH undergoes hierarchical folding upon binding to Rac1. a** $^{15}$N $R_{1\rho}$ relaxation rates (25 °C, 700 MHz) of POSH$_{315-380}$ without (gray) and with Rac1 at 23% (orange), 45% (red), and 76% (blue) molar ratios. **b** $^{15}$N $R_{1\rho}$ relaxation rates (25 °C, 700 MHz) of the CRIB mutant of POSH$_{315-380}$ without (gray) and with Rac1 (45% molar ratio, red). In panel a and b, error bars represent one standard deviation of the relaxation rates obtained from Monte Carlo resampling of intensity decays (see "Methods"). **c** Schematic representation of 3-site and 4-site exchange models used for CEST data analysis. **d** Analysis of $^{15}$N CEST data of POSH with 20% Rac1 using a 3-site model. Experimental data (red circles) were globally analyzed across all residues and all $B_1$ fields (red lines). Selected CEST profiles ($B_1$ = 20.5 Hz) are shown. Vertical gray lines represent the chemical shifts of free POSH (solid line), intermediate B (dash-dotted line) and the final complex (dashed line). Data points shown in pink were excluded from the analysis (see "Methods"). **e** Analysis of $^{15}$N CEST data of POSH with 20% Rac1 using a 4-site model. Vertical gray lines represent

the chemical shifts of free POSH (solid line), intermediate A (dotted line), intermediate B (dash-dotted line) and the final complex (dashed line). In panel d and e, error bars represent uncertainties estimated from noise variance in the CEST profiles (see "Methods"). **f–h** Chemical shift differences between intermediate A and free POSH (**f**, $\Delta\omega_{FA}$), between intermediates B and A (**g**, $\Delta\omega_{AB}$) and between the complex C and intermediate B (**h**, $\Delta\omega_{BC}$). In panels (**f–h**), error bars represent uncertainties estimated from the covariance matrix of the Levenberg-Marquardt minimization. **i** Structure of the POSH-Rac1 complex highlighting the closed switch I loop of Rac1. The van der Waals surfaces of T338 and N336 (POSH) and P34 (Rac1) are shown. **j** Same as in panel (**i**), but with the switch I loop in an open conformation. **k** Schematic of the hierarchical folding trajectory of POSH derived from $^{15}$N CEST data analysis using the 4-site exchange model. Source data are provided as a Source data file.

which POSH fine-tunes signaling outcomes while serving as a platform for assembling multiprotein complexes.

More broadly, our work enhances our understanding of the binding interfaces and the specific interactions that drive extensive folding-upon-binding of IDPs. These insights pave the way for improving structure prediction of IDP complexes, which remain challenging targets due to the low-quality multiple sequence alignments of IDPs. Furthermore, atomic-resolution characterization of hierarchical folding trajectories creates opportunities for drug discovery targeting IDP complexes. Our studies of the POSH-Rac1 complex reveal that IDPs can undergo well-defined conformational transitions upon binding, accompanied by gradual formation of secondary structure. These transitions give rise to intermediate states that, in principle, could be selectively stabilized or perturbed by small molecules or peptides. Structural snapshots of these intermediates, as obtained in this work, thus serve as valuable starting points for therapeutic exploration, offering alternative strategies to modulate IDP function across diverse biological contexts.

## Methods

### Protein expression and purification

Human Rac1 (Uniprot P63000; residues 1–177 omitting the oligomerization-promoting C-terminal tail) and POSH$_{260-445}$ (corresponding to the intrinsically disordered region (residues 260–445) between the second and third SH3 domains of human SH3RF1 Uniprot Q7Z6J0; also known as 'plenty of SH3 domains' or POSH), as well as the Rac1$_{1-177}$–POSH$_{319-371}$ fusion construct were subcloned into a pESPRIT vector[42] containing an N-terminal 6xHis-tag and a tobacco etch virus (TEV) cleavage site, while the shorter POSH$_{315-380}$ construct was subcloned into a pET-28a derived vector with an N-terminal thioredoxin (Trx)-6xHis-tag and a TEV cleavage site for improved expression[43]. The final proteins after protease cleavage contain N-terminal GRR (Rac1$_{1-177}$, POSH$_{260-445}$, and Rac1$_{1-177}$-POSH$_{319-371}$ fusion) or GHMW extensions (POSH$_{315-380}$).

Transformed *E. coli* BL21(DE3) cells were grown in Lysogeny Broth (LB) (unlabeled protein) or M9 minimal medium ($^{15}$N and $^{13}$C$^{15}$N uniformly labeled protein; 1 g/L $^{15}$NH$_4$Cl and 2 g/L $^{13}$C-labeled D-glucose for nitrogen and carbon isotope labeling, respectively) to an optical density at 600 nm (OD$_{600}$) of 0.6–0.8 at 37 °C, induced with isopropyl β-D-1-thiogalactopyranoside (IPTG) (1 mM) and left to express at 20 °C overnight. Cells were harvested by centrifugation (5000 × g, 20 min, 4 °C) and the pellets were stored at −80 °C until purification.

Cell pellets were resuspended in purification buffer (Rac1$_{1-177}$ and Rac1$_{1-177}$–POSH$_{319-371}$ fusion: 50 mM Tris-HCl, 150 mM NaCl, 5 mM MgCl$_2$, 5 mM β-mercaptoethanol (BME) pH 8.0; POSH$_{260-445}$ and POSH$_{315-380}$: 20 mM Tris-HCl, 150 mM NaCl, 5 mM BME pH 8.0) supplemented with protease inhibitors (Roche cOmplete tablet), and the cells were lysed by sonication. The lysate was clarified by centrifugation (18,000 × g, 30 min, 4 °C) and passed over a 0.22 μm filter. Crude protein was purified using immobilized metal ion affinity

chromatography (IMAC). The lysate was incubated with HisPur Ni-NTA resin (Thermo Fischer) for 30 min at 4 °C with gentle agitation before loading onto the column. For POSH$_{315-380}$, 6 M urea was included during the incubation step to enhance binding efficiency. After application, the column was washed with purification buffer supplemented with 7 mM imidazole (3 column volumes) and bound protein was eluted using purification buffer containing 350 mM imidazole. The His-tag was cleaved using TEV protease (2–4 mg/L of culture) during overnight dialysis against purification buffer and removed by a second passing over the IMAC column. For the POSH$_{260-445}$, Rac1$_{1-177}$ and Rac1$_{1-177}$–POSH$_{319-371}$ constructs, the proteins eluted in the flow-through, while POSH$_{315-380}$ eluted with purification buffer with 15 mM imidazole. The POSH$_{260-445}$ construct underwent further purification via ion exchange chromatography: the sample was first passed over an anion exchange column (Q-Sepharose Fast Flow, GE Healthcare) followed by binding to a cation exchange column (CM-cellulose resin, GE Healthcare) and elution with purification buffer containing 150 mM NaCl.

Following IMAC purification, Rac1 was found in the GDP-loaded form and a nucleotide exchange reaction was performed to load the protein with the non-hydrolysable GTP analogue GMPPNP (Guanosine 5′-[β,γ-imido]triphosphate trisodium salt hydrate powder, Sigma Aldrich). Calf intestine alkaline phosphatase beads (Sigma Aldrich) were rinsed four times with bead wash buffer (10 mM Tris-HCl pH 7.5, 0.1 mM ZnCl$_2$) by resuspending the beads in 1 mL buffer followed by centrifugation (1000 × g, 2 min) to settle the beads and to remove the supernatant. After rinsing, the beads were resuspended in nucleotide exchange buffer (3 M (NH$_4$)$_2$SO$_4$, 10 mM Tris-HCl pH 7.5, 91 μL) and GMPPNP (20 mM, 333 μL). Rac1 (500 μM, 500 μL) was added last to the reaction mixture. The reaction mixture was incubated at 37 °C with agitation for 4.5 h. The alkaline phosphatase beads were recovered by centrifugation (1000 × g, 5 min) with the supernatant containing the GMPPNP-loaded protein.

For all constructs, size exclusion chromatography was used as a final step. A Superdex S75 10/300 GL column (Cytiva) was used for the Rac1$_{1-177}$ and Rac1$_{1-177}$–POSH$_{319-371}$ constructs and a Superdex S200 Increase 10/300 GL column (Cytiva) was used for POSH$_{260-445}$ and POSH$_{315-380}$ constructs. The columns were equilibrated with the buffer used for subsequent experiments.

### NMR spectral assignment and secondary structure propensities of POSH

Spectral assignments of the two POSH constructs were obtained at 25 °C using protein samples of 500 μM in 50 mM sodium phosphate buffer pH 6.0, 150 mM NaCl, 5 mM MgCl$_2$, 10 mM dithiothreitol (DTT) with added protease inhibitor (Roche cOmplete) and D$_2$O (7% v/v). A set of BEST-type triple resonance spectra were recorded: HNCO, intra-residue HN(CA)CO, HN(CO)CA, intra-residue HNCA, HN(CO)CACB and intra-residue HNCACB at a $^1$H frequency of 600 MHz for POSH$_{260-445}$ and HNCO, HN(CO)CACB and intra-residue HNCACB at a $^1$H frequency

of 700 MHz for POSH$_{315-380}$[44,45]. Both datasets were recorded on Bruker spectrometers running Topspin version 3.5 and equipped with 5 mm cryogenically cooled probes. The program MARS (version 1.2 Linux) was used for automatic assignment of spin systems followed by manual verification[46]. Backbone resonances of POSH$_{260-445}$ were assigned for 150 of 160 non-proline residues. The resonances of K291, H292, S293, H303, S304, A341, L399, A421, A422, and A423 could not be reliably assigned as these residues are surrounded by prolines or are part of repetitive sequences. For POSH$_{315-380}$ all non-proline residues were assigned. Secondary structure propensities were calculated using the neighbor-corrected secondary propensity calculator using the experimental Cα and Cβ chemical shifts as input[47,48].

## Chemical shift titrations and relaxation measurements

Chemical shift titrations of POSH with GMPPNP-loaded Rac1 were carried out at 25 °C and at a $^1$H frequency of 600 MHz (Bruker, Topspin version 3.5) by acquiring a series of $^1$H-$^{15}$N HSQC spectra of $^{15}$N POSH$_{260-445}$ (200 μM) with 0, 45, and 150% (molar ratio) of Rac1, of $^{15}$N POSH$_{315-380}$ (200 μM) with 0, 23, 45, 76, 100, and 200% (molar ratio) of Rac1 and of the CRIB mutant ($^{326}$ISPP$^{329}$ → $^{326}$RRPR$^{329}$) of $^{15}$N POSH$_{315-380}$ (200 μM) with 0, 76, 150% (molar ratio) of Rac1. $^{15}$N $R_{1\rho}$ relaxation rates were measured at 25 °C and at a $^1$H frequency of 700 MHz (Bruker, Topspin version 3.5) for POSH$_{260-445}$ with 0 and 45% Rac1, for POSH$_{315-380}$ with 0, 23, 45 and 76% Rac1 and for the CRIB mutant of POSH$_{315-380}$ with 0 and 45% Rac1 using HSQC-based pulse sequences[49]. The $^{15}$N $R_{1\rho}$ relaxation experiments were recorded with a spin lock field of 1.5 kHz and an interscan delay of 1.8 s. The magnetization decay was sampled at 1, 10, 30, 50, 70 (×2), 90, 130, 170, 210, and 250 ms. The $R_{1\rho}$ relaxation rates were determined by fitting the time-dependent signal intensities to an exponential decay. Uncertainties in the relaxation rates were estimated using a Monte Carlo approach, where the noise level in each plane (corresponding to each time delay) of the relaxation experiment was used as the measurement error for the signal intensities.

## CEST experiments

The $^{15}$N CEST experiments of POSH$_{260-445}$ were acquired at a $^1$H frequency of 700 MHz (Bruker, Topspin version 3.5) at 25 °C with 0 or 20% molar ratio of Rac1 using the published pulse sequence[50]. A saturation period of 0.4 s and a $B_1$ field strength of 21.6 Hz were used. The dataset comprising 68 two-dimensional planes was acquired by varying the saturation frequency in the range from 104 to 132 ppm with a step size of 30 Hz. The $^{15}$N chemical shift differences between free and Rac1-bound POSH$_{260-445}$ were extracted directly as the difference between the major and minor dips observed in the CEST profiles.

The $^{13}$C′ CEST experiment of POSH$_{315-380}$ was acquired at a $^1$H frequency of 950 MHz (Bruker, Topspin version 4.1.1) with a 20% molar ratio of Rac1 using a HNCO-based pulse sequence[51]. A saturation period of 0.4 s and a $B_1$ field strength of 21.6 Hz were used. The dataset, comprising 86 two-dimensional planes, was acquired by varying the saturation frequency in the range from 171 to 186 ppm with a step size of 40 Hz. The $^{13}$C′ chemical shift differences between free and Rac1-bound POSH$_{315-380}$ were extracted as the difference between the major and minor dips observed in the CEST profiles.

The $^{15}$N CEST experiments of POSH$_{315-380}$ were recorded with a 20% molar ratio of Rac1 and at a $^1$H frequency of 950 MHz (Bruker, Topspin version 4.1.4). To accelerate data acquisition, we employed a DANTE multi-frequency irradiation scheme (D-CEST)[52] for three experiments ($B_1$ = 5.2, 10.3, and 20.5 Hz and with DANTE windows of 240, 448, and 800 Hz) using a modification of the published pulse sequence with a semi-constant time chemical shift evolution in the indirect dimension. The frequency of the saturating field was varied with a step size of 8, 16, and 32 Hz for spectra recorded with windows of 240, 448, and 800 Hz, respectively, and all D-CEST spectra were acquired with a saturation period of 0.4 s. The classical single-frequency irradiation scheme[50] was employed in one experiment

($B_1$ = 68.6 Hz) to facilitate the deconvolution of minor state frequencies. A saturation period of 0.4 s was used and the frequency of the $^{15}$N saturating field was varied in the range from 103 to 134 ppm with a step size of 100 Hz. To eliminate artefacts on the CEST profiles from solvent exchange of the amide protons of POSH, an external D$_2$O lock was employed with a 3 mm NMR tube containing the sample placed inside a 5 mm tube with D$_2$O[53].

## Analysis of the CEST data

The CEST data were analyzed using the program ChemEx (version 2025.4.0)[50] implementing two distinct exchange models. The first model is a linear 3-site exchange model, assuming that POSH transitions from its free state (F) through an intermediate (B) before reaching its final bound complex (C):

$$\text{POSH(F)} + \text{Rac1} \underset{k_{\text{off}}}{\overset{k_{\text{on}}[\text{Rac1}]}{\rightleftarrows}} \text{POSH(B)} : \text{Rac1} \underset{k_{\text{CB}}}{\overset{k_{\text{BC}}}{\rightleftarrows}} \text{POSH(C)} : \text{Rac1}$$

The second model is a linear 4-site exchange model, assuming that POSH transitions through two structurally distinct intermediates (A and B) before adopting its final bound conformation:

$$\text{POSH(F)} + \text{Rac1} \underset{k_{\text{off}}}{\overset{k_{\text{on}}[\text{Rac1}]}{\rightleftarrows}} \text{POSH(A)} : \text{Rac1} \underset{k_{\text{BA}}}{\overset{k_{\text{AB}}}{\rightleftarrows}} \text{POSH(B)} : \text{Rac1} \underset{k_{\text{CB}}}{\overset{k_{\text{BC}}}{\rightleftarrows}} \text{POSH(C)} : \text{Rac1}$$

The linear exchange models are justified by the hierarchical nature of the folding process, as evidenced by mutations in the CRIB motif that completely abolish Rac1 binding (Fig. 3b) and by the crystal structure of the POSH·Rac1 complex, which reveals that MRE2 partly covers MRE1 (Fig. 2g). Uncertainties in the CEST profiles were estimated using the "scatter" option in ChemEx, which is adapted from the "estimatenoise" MATLAB function[54]. This method calculates the noise variance by applying a finite-difference filter that removes the underlying smooth signal component from the CEST profiles. A key advantage of this approach is its ability to robustly estimate the noise level even in profiles with limited baseline regions. To ensure the robustness of the fit, data points with ±$B_1$ (in Hz) of the major state frequency were excluded from the analysis (indicated as pink points in relevant figures). This region is most sensitive to distortions from $B_1$ inhomogeneity and the presence of decoupling sidebands, while being dominated by on-resonance saturation effects that provide minimal information about the chemical exchange process. Uncertainties in the fitted parameters were estimated from the covariance matrix of the Levenberg-Marquardt minimization algorithm implemented in ChemEx.

In a first step, data analysis was carried out in a residue-specific manner across all $B_1$ fields using the 3-site exchange model. Initial values for chemical shift differences (Δω) between major and minor states were determined by manually selecting the minor state frequencies using the D-CEST picking tool in ChemEx. To prevent overfitting, we made two assumptions: (1) the sum of the populations of the intermediate ($p_B$) and the fully bound state ($p_C$) was fixed to that calculated from the protein concentrations ([Rac1] = 100 μM, [POSH] = 500 μM) and the dissociation constant obtained from ITC ($K_d$= 24 μM at 25 °C) and (2) the transverse relaxation rate of the intermediate ($R_{2B}$) was assumed to be equal to that of the bound complex ($R_{2C}$). With these assumptions, the following parameters were optimized for each residue: $k_{\text{off}}$, $k_{\text{BC}}$, $k_{\text{CB}}$, Δω$_{FB}$, Δω$_{FC}$, $R_{1A}$, $R_{2F}$ and $R_{2B}$ (=$R_{2C}$). The goal of this single-residue analysis was to assign the minor states observed in the CEST profiles to either the folding intermediate (B) or the final bound complex (C), with particular focus on residues where two well-separated minor states were detected. Initially, the minor state closest to the major state (free state, F) was assigned to B, while the minor

state furthest from F was assigned to C. This assignment consistently showed that $p_B$ was lower than $p_C$. Swapping the assignments, *i.e.* designating the minor state furthest from the major state as B, either increased residuals or caused the fitting algorithm to revert to the original configuration, thereby supporting our initial assignment strategy. Therefore, we could confidently assign the minor states observed in the CEST profiles to state B or C on the basis of their populations (Supplementary Fig. 9).

In a second step, we proceeded with a global 3-site analysis of the CEST data across all residues in the Rac1-binding region of POSH (residues 325–371) and across the four $B_1$ field strengths, while again fixing the sum of the populations of state B and C to that calculated from the complex dissociation constant. The CEST data of residues I326 and L346 were excluded due to spectral overlap. The following parameters were optimized: $k_{off}$, $k_{BC}$, and $k_{CB}$ (global parameters) and $\Delta\omega_{FB}$, $\Delta\omega_{FC}$, $R_{1F}$, $R_{2F}$, $R_{2B}$ and $R_{2C}$ (residue-specific parameters). For certain residues, particularly those with overlapping minor states (B and C), the transverse relaxation rates of the bound states were assumed to be identical ($R_{2B} = R_{2C}$) to achieve convergence of the fit. This applied to residues 325, 327, 349, 352, 361, 366, and 367.

In the third step of our analysis, we performed a global fit of the experimental CEST data using the 4-site exchange model. The sum of the populations of states A, B, and C was constrained to match that calculated from the measured complex dissociation constant. The following parameters were defined: $k_{ex,AB} = k_{AB} + k_{BA}$, $k_{ex,BC} = k_{BC} + k_{CB}$, $K_{eq,AB} = p_B/p_A$ and $K_{eq,BC} = p_C/p_B$. The parameters optimized in the 4-site global analysis were: $k_{off}$, $k_{ex,AB}$, $k_{ex,BC}$, $K_{eq,AB}$, and $K_{eq,BC}$ (global) and $\Delta\omega_{FA}$, $\Delta\omega_{FB}$, $\Delta\omega_{FC}$, $R_{1F}$, $R_{2F}$, $R_{2B}$ and $R_{2C}$ (residue-specific). The transverse relaxation rate $R_{2A}$ was constrained to that of the free state ($R_{2A} = R_{2F}$) for residues outside the partial CRIB motif and to that of the second folding intermediate ($R_{2A} = R_{2B}$) for residues within the motif (defined as residues 325–333, see below). For parameters shared with the 3-site model, the same initial values were retained. The initial values of the chemical shift difference between the free state and folding intermediate A ($\Delta\omega_{FA}$) were set to $\Delta\omega_{FA} = \Delta\omega_{FB}$ for residues within the partial CRIB motif and to $\Delta\omega_{FA} = 0$ ppm for residues outside the motif. The residue boundaries of the CRIB-anchored state were determined by systematically varying the residue limits within the range 325–335 and performing a global 4-site analysis for each combination. The optimal subset − residues 325, 327, 330, 331, and 333 − yielded the lowest residuals, both within the CRIB motif and across the entire dataset. Incorporating this second folding intermediate (the CRIB-anchored state) led to a marked improvement in the agreement with the experimental data. This improvement was especially pronounced for residues within the partial CRIB motif, but extended across the entire POSH sequence primarily due to a redistribution of the populations of the various states relative to the 3-site model (Supplementary Fig. 12a, b).

We estimated the uncertainties of the globally-fitted exchange parameters using a bootstrap analysis. A total of 300 bootstrapped datasets were generated. Each synthetic dataset was constructed as follows: for every individual experimental CEST profile, a new profile was created by randomly sampling its original data points with replacement, such that the total number of points per profile was retained. This process was repeated for all profiles to create a complete synthetic dataset. Each of the 300 datasets was then fit globally using either the 3-site or 4-site exchange model. The 68% confidence interval of the resulting distribution for each global parameter was reported as its uncertainty.

## Isothermal titration calorimetry
ITC measurements were performed on a MicroCal PEAQ ITC calorimeter (Malvern Instruments). Protein samples were dialyzed against 50 mM HEPES pH 6.9, 150 mM NaCl, 5 mM MgCl$_2$, 2 mM TCEP and the dialysate was used for further dilution of the samples. Duplicate

experiments were performed at 5, 10, 15, and 35 °C with POSH$_{315-380}$ (1000 μM) titrated into GMPPNP-loaded Rac1 (125 μM). Results from three injections into the same cell content of Rac1 were merged by concatenation using the Malvern MicroCal Concat software (3 × 13 injections of 3 μL each with 180-second intervals, stirring speed 750 rpm), and the data were analyzed according to the binding model "One set of sites" using the PEAQ-ITC analysis software. The uncertainty in the thermodynamic parameters was determined from the standard deviation of duplicate titrations. The dissociation constant, $K_d$, could not be determined at 25 °C as the binding enthalpy was close to 0 kcal/mol. Instead, the $K_d$ was estimated through linear extrapolation of the Gibbs free energy measured at the other temperatures ($K_d = 24$ μM at 25 °C).

The binding entropy ($\Delta S$) was dissected into three contributions from conformational changes, $\Delta S_{conformational}$, from desolvation of protein surfaces, $\Delta S_{desolvation}$, and from rotational and translational motions, $\Delta S_{rt}$:

$$\Delta S(T) = \Delta S_{conformational} + \Delta S_{desolvation}(T) + \Delta S_{rt} \quad (1)$$

The contribution from rotational and translational motion is assumed to be a constant, interaction-independent value ($\Delta S_{rt} = -110$ J·mol$^{-1}$·K$^{-1}$). The conformational entropy was determined from an empirical equation originally developed by Spolar and Record[30] and recently re-parameterized for IDP complexes[31]:

$$\Delta S^{\circ}_{conformational} = -1.66\Delta C_p \ln\left(\frac{T_S}{386}K\right) + 110 \text{J} \cdot \text{mol}^{-1} \cdot \text{K}^{-1} \quad (2)$$

where $\Delta C_p$ is the change in heat capacity at constant pressure (derived from the change in $\Delta H$ with temperature) and $T_S$ is the iso-entropic temperature (307.1 K, derived from the change in $\Delta S$ with temperature).

## Crystallization of the Rac1·POSH$_{321-348}$ peptide complex
High-throughput crystallization screening using sitting drop vapor diffusion was performed at the crystallization (HTX) platform hosted by the European Molecular Biology Laboratory (EMBL), Grenoble. Screening was performed with GMPPNP-loaded Rac1 (residues 1–177) at 3–6 mg/mL with 10-fold excess of POSH$_{321-348}$ (lyophilized peptide purchased from CASLO A/S) in 20 mM Tris-HCl pH 8, 150 mM NaCl, 5 mM MgCl$_2$, 1 mM DTT, 1 mM GMPPNP. Six standard crystallization screens (Wizard I + II, Rigaku; Salt-Grid, Hampton; JCSG, Molecular Dimensions; PACT, Molecular Dimensions; PEGs I, Qiagen; and Classics Suite, Qiagen) at 4 and 20 °C were used thereby testing a total of 1152 different conditions. Crystals were harvested by the HTX platform using the CrystalDirect technology[55,56] and diffracted on the MASSIF-1 beamline[57,58] at the European Synchrotron Radiation Facility (ESRF), Grenoble. The best diffraction data were obtained with crystals from 6 mg/mL Rac1 in 0.05 M potassium phosphate and 20% (w/v) PEG 8000 as the precipitant solution (Classics Suite, Qiagen, F9 condition) grown at 4 °C. The structure was solved at 2 Å resolution revealing two Rac1·POSH peptide complexes with 1:1 stoichiometry in the asymmetric unit.

## Crystallization of the Rac1·POSH$_{319-371}$ fusion complex
High-throughput crystallization screening using sitting drop vapor diffusion was performed at the HTX platform with the GMPPNP-loaded Rac1·POSH$_{319-371}$ fusion construct at 4–10 mg/mL in 20 mM Tris-HCl at pH 8, 150 mM NaCl, 5 mM MgCl$_2$, 1 mM DTT, 1 mM GMPPNP. Standard screens (Wizard I + II, Rigaku; Salt-Grid, Hampton; JCSG, Molecular Dimensions; PACT, Molecular Dimensions; PEGs I, Qiagen; and Classics Suite, Qiagen) and screens optimized specifically to promote protein complex formation (Morpheus and ProPlex; Molecular dimensions) at 4 and 20 °C were used thereby testing a total of 1536 different

conditions. Crystals were harvested by the HTX platform and diffracted on the MASSIF-1 beamline at the ESRF. The best diffraction data were obtained with crystals from 2 mg/mL Rac1-POSH$_{319-371}$ fusion in 0.12 M ethylene glycols, 0.1 M imidazole-MES pH 6.5, 37.5% (v/v) 2-methyl-2,4-pentanediol (MPD)-PEG 1000-PEG 3350 as the precipitant solution (Morpheus, Molecular Dimensions, E4 condition) grown at 4 °C. The structure was solved at 1.2 Å resolution revealing one Rac1-POSH fusion complex with 1:1 stoichiometry in the asymmetric unit.

### Data processing and refinement

Crystal diffraction was performed at the ID30A-1/MASSIF-1 beamline (ESRF synchrotron, Grenoble, France) equipped with a Pilatus detector (Dectris). In all cases, data processing was performed with autoPROC[59] (versions 1.0.5 or 1.1.7, Global Phasing Ltd.), including the STARANISO[60] program for anisotropic data analysis.

In the case of Rac1-POSH$_{321-348}$ peptide complex, starting from reflection data generated by the MASSIF-1 automatic processing pipeline, a first model was obtained by performing molecular replacement with Phaser[61]; a published structure of Rac1-GMPPNP (PDB: 3TH5)[62] served as a search model. X-ray diffraction data were subsequently reprocessed using the Pipedream[63] pipeline (version 1.3.1, Global Phasing Ltd.) for molecular replacement and model refinement from several datasets. In particular, for molecular replacement with Phaser we used the initial Rac1-POSH$_{321-348}$ peptide model, which was then refined with BUSTER[64] (version 2.10.4, Global Phasing Ltd). Final cycles of manual model adjustment and refinement were performed with Coot[65] (version 0.9.8.1) and Refmac[66] (version 5.8.0411), respectively.

In the case of the Rac1-POSH$_{319-371}$ fusion complex, phases were found by molecular replacement with Phaser, using the Rac1-GMPPNP structure as a search model; the initial solution was improved through cycles of manual adjusting in Coot (version 0.9.8.1) and refined using Refmac5 (version 5.8.0403). In all cases, the peptide chain of POSH, the Rac1-POSH linker (for the fusion construct), and the switch regions of Rac1 were manually built and/or readjusted. Regions of poor electron density were left unmodelled (these regions were added as random coil chains using pdbfixer for illustrative purposes in all figures). Phaser and Refmac were all used as programs of the CCP4 suite[67] (version 8.0.008 or successive versions) and crystallography applications were compiled and configured by SBGrid[68].

### Native mass spectrometry

Samples of GDP- and GMPPNP-loaded Rac1 (5 μM) in 250 mM ammonium acetate buffer with 10 mM DTT were analyzed by native mass spectrometry (MS)[69]. Samples of Rac1-GDP and Rac1-GMPNP were analyzed, each with one technical replicate. GDP and GMPPNP (in absence of Rac1) were analyzed as control samples. Protein ions were generated using a nanoflow electrospray (nano-ESI) source. Nanoflow platinum-coated borosilicate electrospray capillaries were bought from Thermo Electron SAS (Courtaboeuf, France). MS analyses were carried out on a quadrupole time-of-flight mass spectrometer (Q-TOF Ultima, Waters Corporation, Manchester, UK). The instrument was modified for the detection of high masses[70,71]. The following instrumental parameters were used: capillary voltage = 1.2–1.3 kV, cone potential = 40 V, RF lens-1 potential = 40 V, RF lens-2 potential = 1 V, aperture-1 potential = 0 V, collision energy = 30–140 V, and microchannel plate (MCP) = 1900 V. All mass spectra were calibrated externally using a solution of cesium iodide (6 mg/mL in 50% isopropanol) and were processed using the Masslynx 4.0 (Waters Corporation, Manchester, UK), Massign software package[72] and UniDec[73].

### Reporting summary

Further information on research design is available in the Nature Portfolio Reporting Summary linked to this article.

## Data availability

Protein structure data generated in this study have been deposited in the PDB database with accession codes: 9RFB (Rac1-POSH$_{321-348}$ peptide complex) and 9RFF (Rac1-POSH$_{319-371}$ fusion complex). Protein structure data used in this study can be accessed from the PDB with accession codes: 5N6O (GDP-loaded Rac1) and 3TH5 (GMPPNP-loaded Rac1). The $^1$H, $^{13}$C and $^{15}$N chemical shifts of POSH have been deposited in the Biological Magnetic Resonance Bank with accession codes: 52978 (POSH$_{260-445}$) and 52979 (POSH$_{315-380}$). Native protein mass spectrometry spectra of GDP- and GMPPNP-loaded Rac1 have been deposited as text files in Figshare (https://doi.org/10.6084/m9.figshare.30375364). Source data are provided with this paper.

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

## Acknowledgements

The authors would like to thank Helen Mott and Caroline Mas for fruitful discussions and the ESRF for beamtime access and technical support. This work was funded by the Impulscience® programme of the Fondation Bettencourt Schueller (to M.R.J.) and by the French Agence Nationale de la Recherche (ANR) through project ScaffoldDisorder (ANR-21-CE11-0033, to M.R.J. and A.P.). Financial support is also acknowledged from the Grenoble Alliance for Integrated Structural and Cell Biology (GRAL, Ph.D. fellowship to L.F.K.), from the European Union HORIZON-MSCA-2022-DN-01 funded IDPro doctoral network, grant agreement number 101119633 (Ph.D. fellowship to T.W.), and from the IR INFRA-NALYTICS FR2054. This work used the platforms of the Grenoble Instruct-ERIC center (ISBG; UAR 3518 CNRS-CEA-UGA-EMBL) within the Grenoble Partnership for Structural Biology (PSB), supported by FRISBI (ANR-10-INBS-0005-02) and GRAL, financed within the University Grenoble Alpes graduate school (Ecoles Universitaires de Recherche) CBH-EUR-GS (ANR-17-EURE-0003). The Institut de Biologie Structurale acknowledges integration into the Interdisciplinary Research Institute of Grenoble.

## Author contributions

M.R.J. and A.P. conceived the study. L.F.K., T.W., M.T., and L.M.B. made samples. L.F.K., F.S.I., and A.P. solved the crystal structures of the Rac1-POSH complexes. L.F.K., T.W., E.D., L.M.B., L.M.P., G.B., and M.R.J. designed and performed all NMR experiments. G.B. provided the software for analysis of the CEST data. L.F.K., T.W., E.D., L.M.B., G.B., and M.R.J. analyzed and interpreted the NMR data. L.F.K. carried out the ITC measurements. E.B.E. carried out the native mass spectrometry. L.F.K., M.R.J., and A.P. wrote the manuscript with input from all authors.

## Competing interests

The authors declare no competing interests.
