## [Transparent Peer Review file · Nature Communications]

Hierarchical folding-upon-binding of an intrinsically disordered protein

Corresponding Author: Dr Malene Jensen

Version 0:

Reviewer comments:

Reviewer #1

(Remarks to the Author)

A growing number of structures are now available for intrinsically disordered proteins bound to folded partners, which the community has used to generate several prevailing models to describe these interactions. In most analyzed cases, short (linear) interaction motifs dominate. In a few cases, molecular interaction surfaces have been shown to be surprisingly extensive, yet in these cases it remains rare to observe packing of multiple IDP-derived secondary structure elements with one another in a manner reminiscent of packing in the cores of cooperatively folded proteins. Here the authors present a study of coupled folding and binding of POSH on the GTPase Rac1. Two regions of the investigated POSH disordered region form secondary elements on the surface of Rac1, packing against one another. Furthermore, the reported data supports a hierarchical folding model that is rather unexpected, and which may yield generalizable insights. The detailed analysis of isothermal titration calorimetry data that supports the structural investigation brings together kinetic, thermodynamic, and structural data to yield a complete presentation.

Overall, the data quality in this study appears to be high and the methods used are described with sufficient detail to allow a rigorous interpretation of the results. This manuscript describes an intriguing mechanism for interaction between POSH and Rac1 that encourages the reader to think differently about IDP interactions with folded partners, relative to the standard models that are, to date, more prevalent in the literature. Given the potential significance of this study, it is worth presenting it with the highest clarity. The several suggestions below are motivated by this goal.

1. The sentence on page 5 reading “Notably, most peaks corresponding to the bound state of POSH remain undetectable in the NMR spectra, most likely due to its persistent dynamic nature on the surface of Rac1” could be edited for clarity. Extended Figure 2c displays many (arguably most) resonances in the bound state. I agree with the statement that most resonances associated with direct/specific binding have disappeared, presumably corresponding to MRE1 and MRE2, but most of the resonances are detectable.
2. Page 5 goes on to state that “These data reveal that Rac1 surprisingly engages an extensive 50-residue region of POSH.” Why is that surprising? Extended Figure 1f displays seven different structures of IDPs bound to Rac3 or Cdc42 and, in all cases, IDP segments that appear to be significantly greater than 15 residue SLiMs are implied by ordering in the bound structures. For context, it might be helpful to give the reader an idea of the residue counts included in the ordered regions of these other IDPs. The comparison has the caveat that equivalent solution dynamics data to those presented in the current study may not be available to back up the quantification, but at least it would help to establish the extent of content in comparisons a bit more firmly.
3. On page 7, when discussing the structural features of the POSH-Rac1 complex, the authors state that “These structural insights indicate that POSH stabilizes Rac1 in an active conformation and that MRE2 binding depends on the prior folding of MRE1.” This ordered folding mechanism is a reasonable hypothesis, but it cannot be conclusively established from an endpoint structure alone. At this point in the paper, no kinetic/dynamic evidence has been provided to support this model and refute that the MRE1 and MRE2 regions fold concurrently (in an apparent two-state mechanism).
4. On page 8, when describing NMR intensity mapping in the wild-type and MRE1 mutant complexes, the manuscript asserts that “complex formation is initiated through the partial CRIB motif and that MRE2 cannot bind independently prior to MRE1 folding.” Again, it is not clear that the data included are sufficient to support a sequential folding model, or that they

can exclude the hypothesis that contacts mediated by the mutated residues are necessary to support the apparent two-state folding of the combined MRE1 and MRE2 regions. In contrast, the CEST data described next, along with the presence of two resonances for some residues in the foundational HSQC spectra, do begin to differentiate between these two models. The intensity ratio data are very interesting and appear to be of high quality, so this is not a critique of the data themselves, but only a suggestion that it might be preferable to rephrase the manuscript and build a rhetorical case that does not appear biased toward a desired outcome (i.e., the one that was known by the time of writing).

Reviewer #2

(Remarks to the Author)

This is a beautiful piece of work on an important problem, the interactions between IDPs or IDRs and their targets. The lack of success of the AlphaFold predictions underscores the significance of the work, as pointed out by the authors. The combination of crystallography with NMR exchange and relaxation experiments provides profound insight into not only what the complex looks like, but also how it forms. The ID-adapted SR thermodynamic analysis, based on the calorimetric data was also very nice, combining structure, dynamics and thermodynamics in a very complete package. The exhaustive analysis of the CEST data obtained at multiple fields to resolve, convincingly, the 4-state binding model is very rigorous. Also highly rigorous and convincing is the comparison of the 2 crystal structures, the Rac1/full POSH with that containing only MRE1. Honestly, I do not have any issues whatsoever with the work. I wonder though, if there are other complexes involving CRIB motifs for which this mechanism can be inferred (or hypothesized) from sequence comparisons or intermolecular covariance analysis.

Reviewer #3

(Remarks to the Author)

This manuscript presents an interesting conformational transition as the IDP-containing POSH with little solution structure binds to its cognate small GTPase, Rac1. The structural and biophysical work is extensive, rigorous and intriguing, revealing a two-step concerted process that leads to folding of MRE1 and MRE2 motifs onto Rac1. However, the work lacks a functional context - it is unclear how this hierarchical pathway and the structure of Rac1/POSH relates to its role in cell signaling.

Below are more specific questions regarding the manuscript that should be addressed.

1. Concerning the crystalline structure: the MRE2 motif is not in the peptide, presumably a short peptide was synthesized for practical purposes (authors should explain why a shorter peptide relative to fusion was used). The crystal structures can only be compared for the MRE1 region (Ext Fig 4b), to verify no structural differences due to the fusion construct. Therefore, strictly speaking from structural data, authors cannot rule out the possibility that the beta-hairpin of MRE2 is a consequence of the fusion of the POSH region 319-371 and Rac1. Could the authors respond to this limitation from their data. In other words, can supporting data for the beta-hairpin be provided from in vitro experiments, cellular data, eg mutagenesis/cell signaling? If the authors had used a peptide or recombinant construct spanning 318-371, crystallized the complex, and it matched the fusion construct, that would be supportive. What other evidence is there that the structure is physiologically relevant? Note that NMR relaxation (315380) and 15N-CEST is not direct evidence for this beta-hairpin structure - they only provide evidence for a multi-step folding process.

2. Extending the above question, what is the physiological relevance of this intriguing hierarchical folding pathway? Is the MRE1 motif sufficient for function in cells? What is the effect of truncating the recognition site and dispensing the MRE2 (beta-hairpin) motif? The authors noted that MRE2 partially overlaps with MRE1, and there are side-chain specific contacts. What if these are mutated, what is the effect on Rac1/POSH complex formation in cells?

In summary the manuscript would benefit to a shift in focus away from IDPs, as that shift would impact the wider scientific community. A few papers from 2003-10 (Refs 27-29) are cited regarding POSH/Rac1 signaling in cells. However, it is important to understand how this novel structural work impacts the current state of knowledge about Rac and Cdc42 pathways, JNK signaling, etc. Currently the abstract and final discussion on the generality of this structure to IDPs is nebulous. The composition, length, structure/dynamics and function of IDPs is vast and concluding general principles in this context provides no significant insight. For example, the authors discuss folding intermediates that could be druggable. Which intermediates? And how could they be targeted in a practical way from the current structural work?

The more interesting questions relate to the relevance of hierarchical folding of CRIB like motifs generally in Cdc42-family proteins and cell signaling. Do some or all of the signaling complexes follow a similar folding upon binding behaviour, and what could be the functional relevance?

Typos, cosmetic issues, minor suggestions

- space in word 'discernible', top sentence on page

- ext fig 4a - impossible for readers to compare superposed ribbon models on surface, too small and green/yellow colours difficult to distinguish. I would recommend close-ups only and contrasting color selection, perhaps making surface transparent as it is not critical

- abstract is 'generic' and vague, in the sense that no specific protein is mentioned at all, readers are given no clues as to the

specific pathways (Rac/Cdc42), biological significance, etc. Details must be provided regarding which IDP and signaling complexes as these are essential elements for the readership to link to their own research.

- why was a 45% POSH molar ratio, relative to Rac1, used in the chemical shift assays? What is the stoichiometry of the peptide POSH complex with Rac1 in crystals, is it 1:1, same as solution? The ITC clearly indicates 1:1 complexes, but Ext. Table 2 does not provide sufficient information. Presumably the fusion crystallizes as 1:1 due to covalent link but details should be provided for the peptide/Rac1. The tetragonal crystals appear to have 2 molecular complexes in the asymmetric unit (twice number of atoms).

- what is the buried surface area, van der Waals, number of H-bonds/ salt bridges, etc? The involvement of 50 residues in complex formation is huge. Cdc42 and Par6 are nM affinity, very strong, and my guess with others in Ext Fig 4A (PAK6, WASP, etc) is that they are similarly high nanoMolar affinity. How come the affinity of POSH/Rac1 is so weak? That seems very unusual. These issues are important as they directly relate to functional effects in cells.

- Fig 2d - what is an 'unbiased' electron density map (Fo-Fc)? Using Fc, implying the map arises from a molecular replacement model, means that the map by definition is biased with phases initially from the MR model 3TH5. A more unbiased map such as Fo-Fo requires control data set collection with isomorphous crystals of Rac1 only (no peptide). I would suggest supplying composite omit maps to show the electron density of peptide. Data stats and refinement appear excellent, this is a relatively trivial matter but it is important to be correct regarding model quality.

Version 1:

Reviewer comments:

Reviewer #1

(Remarks to the Author)

The authors have done a nice job of addressing the questions I raised in the prior round of reviews. I have no further suggestions for revision.

Reviewer #3

(Remarks to the Author)

This manuscript is an interesting study of the stepwise folding pathway of an IDP effector of Rac1. The authors have positively responded to my suggestions in several ways. The names of the proteins being studied are now included in the abstract. Additional ¹³C chemical shifts have been collected, supporting the beta-strand conformations that are seen in the crystal structure of the Rac1-POSH fusion protein. In vitro mutagenesis support the idea that MRE1 is required for MRE2 folding, and the authors have also extended their discussions to other CRIB-containing effectors. However, the physiological relevance of this intriguing IDP effector folding pathway has not been addressed. In addition, why the affinity is so weak remains unknown although the authors raise several hypotheses. These important questions remain to be addressed in future studies.

Reviewer #1:

A growing number of structures are now available for intrinsically disordered proteins bound to folded partners, which the community has used to generate several prevailing models to describe these interactions. In most analyzed cases, short (linear) interaction motifs dominate. In a few cases, molecular interaction surfaces have been shown to be surprisingly extensive, yet in these cases it remains rare to observe packing of multiple IDP-derived secondary structure elements with one another in a manner reminiscent of packing in the cores of cooperatively folded proteins. Here the authors present a study of coupled folding and binding of POSH on the GTPase Rac1. Two regions of the investigated POSH disordered region form secondary elements on the surface of Rac1, packing against one another. Furthermore, the reported data supports a hierarchical folding model that is rather unexpected, and which may yield generalizable insights. The detailed analysis of isothermal titration calorimetry data that supports the structural investigation brings together kinetic, thermodynamic, and structural data to yield a complete presentation.

Overall, the data quality in this study appears to be high and the methods used are described with sufficient detail to allow a rigorous interpretation of the results. This manuscript describes an intriguing mechanism for interaction between POSH and Rac1 that encourages the reader to think differently about IDP interactions with folded partners, relative to the standard models that are, to date, more prevalent in the literature.

We thank the reviewer for the positive comments on our manuscript and for highlighting the novelty of our work.

Given the potential significance of this study, it is worth presenting it with the highest clarity. The several suggestions below are motivated by this goal.

1. The sentence on page 5 reading “Notably, most peaks corresponding to the bound state of POSH remain undetectable in the NMR spectra, most likely due to its persistent dynamic nature on the surface of Rac1” could be edited for clarity. Extended Figure 2c displays many (arguably most) resonances in the bound state. I agree with the statement that most resonances associated with direct/specific binding have disappeared, presumably corresponding to MRE1 and MRE2, but most of the resonances are detectable.

We thank the reviewer for highlighting the inaccurate formulation in this sentence. We have revised the phrasing on page 5 of the manuscript to clarify that we are referring to peaks corresponding to residues within the direct interaction site of POSH:

“Notably, most peaks corresponding to residues within the direct interaction site of POSH disappear, however, no new peaks appear in the NMR spectra corresponding to the Rac1-bound state of POSH. This is most likely due to POSH remaining dynamic on the surface of Rac1.”

2. Page 5 goes on to state that “These data reveal that Rac1 surprisingly engages an extensive 50-residue region of POSH.” Why is that surprising? Extended Figure 1f displays seven different structures of IDPs bound to Rac3 or Cdc42 and, in all cases, IDP segments that appear to be significantly greater than 15 residue SLiMs are implied by ordering in the bound structures. For context, it might be helpful to give the reader an idea of the residue counts included in the ordered regions of these other IDPs. The comparison has the caveat that equivalent solution dynamics data to those presented in the current study may not be available to back up the quantification, but at least it would help to establish the extent of content in comparisons a bit more firmly.

We thank the reviewer for this suggestion. In response, we have revised Extended Figure 1f (now Supplementary Figure 1f) to indicate the total number of effector residues observed in previously solved effector–GTPase complex structures. Among these, the shortest effector is PAK1 (22 amino acids) and the longest is WASP (59 amino acids). We agree with the reviewer that these data do not support describing POSH as having an exceptionally long interaction site, and we have accordingly removed the word “surprisingly” from the manuscript text.

3. On page 7, when discussing the structural features of the POSH-Rac1 complex, the authors state that “These structural insights indicate that POSH stabilizes Rac1 in an active conformation and that MRE2 binding depends on the prior folding of MRE1.” This ordered folding mechanism is a reasonable hypothesis, but it cannot be conclusively established from an endpoint structure alone. At this point in the paper, no kinetic/dynamic evidence has been provided to support this model and refute that the MRE1 and MRE2 regions fold concurrently (in an apparent two-state mechanism).

We agree that a static structure cannot definitively establish the order of folding events and that our original wording may have overstated the conclusion. To address this, we have revised the sentence to clarify that our interpretation is consistent with either concurrent or sequential folding. The text now reads:

“These structural insights indicate that POSH stabilizes Rac1 in an active conformation and suggest that MRE2 binding requires MRE1 to adopt a folded state, with binding occurring either concurrently with, or subsequent to, MRE1 folding.”

This wording emphasizes that the crystal structure supports a requirement for MRE1 folding, but does not exclude the possibility of concurrent folding and binding of MRE1 and MRE2.

4. On page 8, when describing NMR intensity mapping in the wild-type and MRE1 mutant complexes, the manuscript asserts that “complex formation is initiated through the partial CRIB motif and that MRE2 cannot bind independently prior to MRE1 folding.” Again, it is not clear that the data included are sufficient to support a sequential folding model, or that they can exclude the hypothesis that contacts mediated by the mutated residues are necessary to support the apparent two-state folding of the combined MRE1 and MRE2 regions. In contrast, the CEST data described next, along with the presence of two resonances for some residues in the foundational HSQC spectra, do begin to differentiate between these two models. The intensity ratio data are very interesting and appear to be of high quality, so this is not a critique of the data themselves, but only a suggestion that it might be preferable to rephrase the manuscript and build a rhetorical case that does not appear biased toward a desired outcome (i.e., the one that was known by the time of writing).

We agree that the mutational experiment and the associated NMR data cannot definitively establish a sequential folding mechanism or exclude alternative folding pathways. To address this, we have revised the text to present our results more cautiously, emphasizing that the data are compatible with hierarchical folding, but do not distinguish whether MRE1 and MRE2 fold sequentially or concomitantly. The revised text now reads (page 9):

“Mutation of the partial CRIB motif (³²⁶ISPP³²⁹ → ³²⁶RRPR³²⁹) abolishes binding of all elements (Fig. 3b, Supplementary Fig. 8c), demonstrating that this motif is essential for binding and that MRE2 cannot interact with Rac1 independently, suggesting that MRE1 folding is required for MRE2 engagement. This observation is in agreement with our crystal structure, which shows MRE2 partially packing on top of MRE1 (Fig. 2g). Collectively, these results are compatible with a hierarchical folding mechanism of POSH, although they do not distinguish between sequential and concomitant folding of MRE1 and MRE2.”

This phrasing avoids overinterpreting the mutational data and clearly motivates the subsequent CEST experiments, which provide additional information on folding kinetics and intermediate states.

Reviewer #2:

This is a beautiful piece of work on an important problem, the interactions between IDPs or IDRs and their targets. The lack of success of the Alphafold predictions underscores the significance of the work, as pointed out by the authors. The combination of crystallography with NMR exchange and relaxation experiments provides profound insight into not only what the complex looks like, but also how it forms. The ID-adapted SR thermodynamic analysis, based on the calorimetric data was also very nice, combining structure, dynamics and thermodynamics in a very complete package. The exhaustive analysis of the CEST data obtained at multiple fields to resolve, convincingly, the 4-state binding model is very rigorous. Also highly rigorous and convincing is the comparison of the 2 crystal structures, the Rac1/full POSH with that containing only MRE1. Honestly, I do not have any issues whatsoever with the work. I wonder though, if there are other complexes involving CRIB motifs for which this mechanism can be inferred (or hypothesized) from sequence comparisons or intermolecular covariance analysis.

We thank the reviewer for the positive evaluation of our manuscript and for recognizing the significance of our findings. We agree that it would be highly interesting to determine whether the hierarchical folding mechanism we describe for the POSH-Rac1 complex is more broadly applicable to other CRIB-containing complexes, and potentially to IDP complexes in general. Although a full investigation of this question lies beyond the scope of the present study, we have expanded the Discussion to address the possible generality of this mechanism, including reference to other GTPase-effector complexes where “dock-and-coalesce” mechanisms have been suggested.

Reviewer #3:

This manuscript presents an interesting conformational transition as the IDP-containing POSH with little solution structure binds to its cognate small GTPase, Rac1. The structural and biophysical work is extensive, rigorous and intriguing, revealing a two-step concerted process that leads to folding of MRE1 and MRE2 motifs onto Rac1. However, the work lacks a functional context - it is unclear how this hierarchical pathway and the structure of Rac1/POSH relates to its role in cell signaling.

Below are more specific questions regarding the manuscript that should be addressed.

1. Concerning the crystalline structure: the MRE2 motif is not in the peptide, presumably a short peptide was synthesized for practical purposes (authors should explain why a shorter peptide relative to fusion was used). The crystal structures can only be compared for the MRE1 region (Ext Fig 4b), to verify no structural differences due to the fusion construct. Therefore, strictly speaking from structural data, authors cannot rule out the possibility that the beta-hairpin of MRE2 is a consequence of the fusion of the POSH region 319-371 and Rac1. Could the authors respond to this limitation from their data. In other words, can supporting data for the beta-hairpin be provided from in vitro experiments, cellular data, eg mutagenesis/cell signaling? If the authors had used a peptide or recombinant construct spanning 318-371, crystallized the complex, and it matched the fusion construct, that would be supportive. What other evidence is there that the structure is physiologically relevant? Note that NMR relaxation (315-380) and 15N-CEST is not direct evidence for this b-hairpin structure - they only provide evidence for a multi-step folding process.

We thank the reviewer for raising this point, which allows us to clarify our crystallization strategy. We were unable to obtain a synthetic peptide comprising both MRE1 and MRE2, as the peptide failed to be produced by chemical synthesis. Furthermore, the POSH construct spanning residues 315–380, expressed in bacteria, did not yield crystals with Rac1 despite extensive trials, likely due to the dynamics of the complex and its relatively low affinity. To overcome this and obtain the structure of Rac1 in complex with POSH encompassing both MRE1 and MRE2, we employed a fusion strategy, linking the N-terminus of POSH to the C-terminus of Rac1 using a linker derived directly from the disordered sequence of POSH preceding the N-terminus of MRE1.

We acknowledge that this fusion could potentially affect the structure of POSH, particularly if the linker were too short to allow proper folding and interactions. To validate the fusion structure, we obtained a structure of Rac1 in complex with a peptide corresponding to MRE1. This region is most relevant for assessing potential fusion artefacts, as the N-terminus of POSH (where MRE1 begins) was fused to the C-terminus of Rac1. If the fusion had altered the conformation of POSH in the complex, it would be detectable by comparing the binding mode of MRE1 in the fusion and peptide complexes. We observe essentially identical binding modes, strongly supporting that the fusion does not perturb the structure of POSH in the complex. MRE2 is located at the C-terminus of POSH and distant from the fusion site and is therefore unlikely to adopt a different structure in an unfused peptide.

To further support the β -strand conformation of MRE2 in solution, we have acquired new carbonyl $^{13}\text{C}'$ CEST experiments. The $^{13}\text{C}'$ chemical shifts are very sensitive to the presence of secondary structure. From these experiments, we have extracted the $^{13}\text{C}'$ chemical shift differences between the free and Rac1-bound form of POSH, which demonstrate that MRE2 adopts β -strand conformations in the complex with Rac1 in solution. These data are now included in Supplementary Figure 3 and referred to on page 5.

2. Extending the above question, what is the physiological relevance of this intriguing hierarchical folding pathway? Is the MRE1 motif sufficient for function in cells? What is the effect of truncating the recognition site and dispensing the MRE2 (beta-hairpin) motif? The authors noted that MRE2 partially overlaps with MRE1, and there are side-chain specific contacts. What if these are mutated, what is the effect on Rac1/POSH complex formation in cells?

We thank the reviewer for their comment regarding the potential functional context of POSH-

Rac1 binding. While we agree that understanding the role of this hierarchical folding mechanism in signaling would be interesting, the focus of the current study is on the structural and mechanistic characterization of IDP interactions, highlighting a new folding-upon-binding mechanism not previously described for IDPs. Accordingly, we have not included functional experiments. We have kept the Discussion focused on the biophysical insights into folding and binding, however, expanding it to address the extent to which this mechanism may be conserved among other GTPase-effector complexes. We believe this focus highlights the mechanistic novelty of the work, while remaining within the intended scope of the study.

In summary the manuscript would benefit to a shift in focus away from IDPs, as that shift would impact the wider scientific community. A few papers from 2003-10 (Refs 27-29) are cited regarding POSH/Rac1 signaling in cells. However, It is important to understand how this novel structural work impacts the current state of knowledge about Rac and Cdc42 pathways, JNK signaling, etc. Currently the abstract and final discussion on the generality of this structure to IDPs is nebulous. The composition, length, structure/dynamics and function of IDPs is vast and concluding general principles in this context provides no significant insight. For example, the authors discuss folding intermediates that could be druggable. Which intermediates? And how could they be targeted in a practical way from the current structural work?

In the revised manuscript, we have expanded the Discussion to include how the hierarchical folding mechanism we describe may be conserved across other GTPase-effector complexes. In addition, we highlight how the POSH-Rac1 interaction distinguishes itself from other GTPase-effector complexes, both in terms of its rather low affinity and its functional role in scaffolding JNK signaling.

We thank the reviewer for raising an important point regarding how conformational transitions and intermediates could be practically targeted. Our intent was not to imply that all folding intermediates of IDPs are inherently druggable, but rather to highlight that structural snapshots of such transitions can provide valuable starting points for therapeutic exploration. In our system, we observe well-defined conformational transitions that involve gradual formation of secondary structure. These transitions give rise to intermediate states that, in principle, could be selectively stabilized or perturbed by small molecules or peptides, thereby modulating IDP function. We have clarified and expanded the Discussion in the revised manuscript to more explicitly convey this point.

The more interesting questions relate to the relevance of hierarchical folding of CRIB like motifs generally in Cdc42-family proteins and cell signaling. Do some or all of the signaling complexes follow a similar folding upon binding behaviour, and what could be the functional relevance?

We thank the reviewer for raising this important point. In the Discussion, we have now considered the extent to which folding-upon-binding occurs in other GTPase-effector complexes, particularly in relation to previous studies describing a “dock-and-coalesce” binding mechanism for GTPase-effector complexes.

Typos, cosmetic issues, minor suggestions:

- space in word ‘discernible’, top sentence on page

This has been corrected.

- ext fig 4a - impossible for readers to compare superposed ribbon models on surface, too small and green/yellow colours difficult to distinguish. I would recommend close-ups only and contrasting color selection, perhaps making surface transparent as it is not critical

We have split the previous Extended Data Figure 4 into three figures (now Supplementary Figures 5, 6, and 7) in order to make each figure panel bigger. We have also changed the yellow color to bright orange to increase contrast with the green color.

- abstract is 'generic' and vague, in the sense that no specific protein is mentioned at all, readers are given no clues as to the specific pathways (Rac/Cdc42), biological significance, etc. Details must be provided regarding which IDP and signaling complexes as these are essential elements for the readership to link to their own research.

We thank the reviewer for this suggestion. We have now mentioned the specific proteins under investigation in the abstract.

- why was a 45% POSH molar ratio, relative to Rac1, used in the chemical shift assays?

All CEST experiments of POSH probing the chemical shift difference between the free and Rac1-bound states were performed with a 20% molar ratio of Rac1 (Figure 1g, Figure 3d,e). This ratio was chosen to enhance the signals of the minor bound state in the CEST profiles, while maintaining a sufficient signal-to-noise ratio for high-quality data acquisition. For the ^{15}N $R_{1\rho}$ relaxation measurements of POSH, a 45% molar ratio of Rac1 was indeed used (Figure 1f). The higher molar ratio increased the $R_{1\rho}$ difference between free POSH and Rac1-bound POSH, thereby allowing us to better define the location of the two molecular recognition elements.

What is the stoichiometry of the peptide POSH complex with Rac1 in crystals, is it 1:1, same as solution? The ITC clearly indicates 1:1 complexes, but Ext. Table 2 does not provide sufficient information. Presumably the fusion crystallizes as 1:1 due to covalent link but details should be provided for the peptide/Rac1. The tetragonal crystals appear to have 2 molecular complexes in the asymmetric unit (twice number of atoms).

Both the Rac1-POSH fusion and the Rac1-POSH peptide crystallized as 1:1 complexes. The fusion construct contained a single complex in the asymmetric unit, whereas the peptide complex contained two. This information is now included in the Methods section (page 23).

- what is the buried surface area, van der Waals, number of H-bonds/ salt bridges, etc? The involvement of 50 residues in complex formation is huge. Cdc42 and Par6 are nM affinity, very strong, and my guess with others in Ext Fig 4A (PAK6, WASP, etc) is that they are similarly high nanoMolar affinity. How come the affinity of POSH/Rac1 is so weak? That seems very unusual. These issues are important as they directly relate to functional effects in cells.

We agree that the extent of interface burial and the number of specific contacts (hydrogen bonds and salt bridges) are important determinants of binding affinity. In the POSH-Rac1 structure, the buried surface area is 1830 Å², with 25 hydrogen bonds stabilizing the interface; no salt bridges are present. These values have now been included in the manuscript in the description of the POSH-Rac1 crystal structure (pages 6–7).

As noted by the reviewer, the micromolar affinity of the POSH–Rac1 complex is relatively weak compared to nanomolar complexes such as Cdc42–Par6, Cdc42–ACK, Cdc42–PAK, and Cdc42–WASP. This difference does not appear to stem from a lack of specific contacts, but rather from the non-canonical nature of the POSH CRIB motif. Previous studies have shown that mutations of the hydrophobic positions following the ISXP motif of the CRIB motif reduce binding up to 100-fold. The CRIB motif of POSH do not have these hydrophobic positions, potentially explaining the lower affinity of the POSH-Rac1 complex.

In addition to this, the extensive folding-upon-binding of POSH imposes a significant conformational entropy penalty. This unfavorable term is only partially compensated by the favorable entropy arising from the release of ordered water molecules from the protein-protein interface upon complex formation. Thus, although ~50 residues are directly involved in binding, the net free energy reflects the interplay between enthalpic stabilization (hydrogen bonds, salt bridges, van der Waals contacts) and these entropic contributions. It is therefore possible that the other GTPase-effector complexes show smaller entropic penalties upon binding and more favorable enthalpic contributions due to the canonical nature of their CRIB motifs, leading to higher affinity. These considerations are now included in the Discussion.

- Fig 2d - what is an 'unbiased' electron density map (Fo-Fc)? Using Fc, implying the map arises from a molecular replacement model, means that the map by definition is biased with phases initially from the MR model 3TH5. A more unbiased map such as Fo-Fo requires control data set collection with isomorphous crystals of Rac1 only (no peptide). I would suggest supplying composite omit maps to show the electron density of peptide. Data stats and refinement appear excellent, this is a relatively trivial matter but it is important to be correct regarding model quality.

We have followed the advice of the reviewer, and we now provide instead the composite omit maps contoured at 1.2σ for both MRE1 and MRE2 in Figure 2.

Reviewer #1:

The authors have done a nice job of addressing the questions I raised in the prior round of reviews. I have no further suggestions for revision.

Reviewer #3:

This manuscript is an interesting study of the stepwise folding pathway of an IDP effector of Rac1. The authors have positively responded to my suggestions in several ways. The names of the proteins being studied are now included in the abstract. Additional ¹³C chemical shifts have been collected, supporting the beta-strand conformations that are seen in the crystal structure of the Rac1-POSH fusion protein. In vitro mutagenesis support the idea that MRE1 is required for MRE2 folding, and the authors have also extended their discussions to other CRIB-containing effectors. However, the physiological relevance of this intriguing IDP effector folding pathway has not been addressed. In addition, why the affinity is so weak remains unknown although the authors raise several hypotheses. These important questions remain to be addressed in future studies.

We thank both reviewers for their comments on our manuscript and for accepting our work for publication in Nature Communications.